# Alternative splicing in seasonal plasticity and the potential for adaptation to environmental change

Rachel A. Steward [1]✉, Maaike A. de Jong[2], Vicencio Oostra[3,4] & Christopher W. Wheat[1,4]✉

Seasonal plasticity is accomplished via tightly regulated developmental cascades that translate environmental cues into trait changes. Little is known about how alternative splicing and other posttranscriptional molecular mechanisms contribute to plasticity or how these mechanisms impact how plasticity evolves. Here, we use transcriptomic and genomic data from the butterfly *Bicyclus anynana*, a model system for seasonal plasticity, to compare the extent of differential expression and splicing and test how these axes of transcriptional plasticity differ in their potential for evolutionary change. Between seasonal morphs, we find that differential splicing affects a smaller but functionally unique set of genes compared to differential expression. Further, we find strong support for the novel hypothesis that spliced genes are more susceptible than differentially expressed genes to erosion of genetic variation due to selection on seasonal plasticity. Our results suggest that splicing plasticity is especially likely to experience genetic constraints that could affect the potential of wild populations to respond to rapidly changing environments.

[1] Zoology Department, Stockholm University, Stockholm, Sweden. [2] Netherlands eScience Centre, Amsterdam, the Netherlands. [3] Institute of Infection, Veterinary and Ecological Sciences, University of Liverpool, Liverpool, UK. [4] These authors contributed equally: Vicencio Oostra, Christopher W Wheat. ✉email: rachel.steward@zoologi.su.se; chris.wheat@zoologi.su.se

Phenotypic plasticity is a widespread adaptation to habitats where conditions fluctuate, because it allows for a single genotype to produce different phenotypes depending on the environment[1,2]. This is especially common in seasonal environments, where large yet predictable resource fluctuations and tight coupling between inductive cues and selective environments have driven the evolution of plasticity across diverse taxa in both temperate and tropical habitats[3,4]. In order to understand how plasticity evolves, it is critical to clarify its underlying molecular mechanisms[5,6]. Plasticity is accomplished via tightly regulated developmental cascades, which translate environmental cues into appropriate trait changes, often across multiple integrated traits[7,8]. While whole gene expression differences are a crucial part of this cascade, alternative splicing and other forms of post-transcriptional regulation are important but poorly characterized additional components that can affect both the amount and type of transcript products from a given locus.

Alternative splicing, in particular, has been predicted as a key molecular mechanism in the production and maintenance of phenotypic plasticity[9]. By combining different whole or partial exons from a single locus into mature transcripts, alternative splicing can increase proteome diversity with the potential to contribute to alternative, even novel, phenotypes while avoiding pleiotropic costs[10,11]. Many case studies support the predicted importance of splicing, with alternative isoform expression of specific candidate genes contributing to or associated with phenotypically plastic traits (e.g.,[12–16]). Despite the critical importance of clarifying molecular mechanisms underlying phenotypic plasticity for understanding its evolution[5,6], little is known about the role of alternative splicing in adaptive plasticity and how it relates to evolvability.

Only recently have studies of plasticity begun to scale up from candidate genes to genome-wide patterns of differential splicing, including asexuality and wing dimorphism in aphids[17], eusocial insect castes[18], temperature adaptation in fruit flies[19,20] and fish[21], and stress responses in flies[22], doves[23], *Daphnia*[24] and minnows[25]. These studies have started to paint a picture of splicing relative to whole gene expression: differential splicing affects a smaller number of genes, with the largest splicing variation often localized in genes that are not differentially expressed between conditions[17,21] (also in sexual dimorphisms[26,27]). For seasonal plasticity, however, the extent to which alternative splicing contributes to transcriptomic variation independently of whole gene expression remains unclear. Further, almost nothing is known about whether these forms of transcript production differ in evolutionary potential.

Clarifying the role of alternative splicing in seasonal plasticity is especially urgent for understanding the potential of natural populations to respond to rapid environmental change. With altered climate regimes and shifting phenologies, seasonal plasticity will need to evolve[28]. Crucially, rapid evolution depends on standing genetic variation for plastic responses[29]. But, in many cases of seasonal plasticity (e.g., diapause in multivoltine species), selection for matching and maintaining optimal responses to seasonal variation can reduce such variation[1,30]. Thus, local adaptation to seasonal climates may leave populations vulnerable to current rapid shifts in seasonal patterns.

Loss of genetic variation through positive or purifying selection is a particular risk for genes that are alternatively spliced. Whole gene expression and alternative splicing are both affected by the intricate interactions of *trans*-acting factors, such as transcription factors that can regulate the expression of multiple target loci throughout the genome, and *cis*-acting regulatory motifs that are physically proximate to the loci they regulate. Although alternative splicing depends on *trans*-acting splicing factors, it is also regulated by enhancer and silencer sequences flanking exon-intron boundaries[31,32], meaning that selection on splicing variation can directly target sites within genes. This type of *cis*-regulation has been shown to affect >50% of splicing variation[33–36], although this varies among species, populations, and splice event types[33,34,37]. Similarly, *cis*-regulatory motifs affecting whole gene expression can be under strong selection[38,39]. However, motifs affecting expression plasticity (e.g., enhancers) are commonly located far enough away from gene bodies that positive or purifying selection on these elements can have little effect on genetic variation within protein-coding regions of the gene, due to linkage decay[40]. Thus, alternatively spliced genes may be more susceptible to erosion of genetic variation compared to differentially expressed genes in large gene networks. To understand the evolutionary importance of differential splicing, especially in the context of environmental change, it is important to quantify whether any such erosion of natural genetic variation occurs in genes where splicing is important for plasticity.

To address these gaps, we assess the role and adaptive potential of splicing in plasticity, using the African butterfly *Bicyclus anynana*, a model for seasonal polyphenism[41]. Its wet and dry season morphs are end points of alternative developmental pathways induced by seasonal temperature variation[42,43], and comprise distinct wing patterns, behaviours and life history strategies (e.g., pace of life, reproductive investment)[44–46]. In these butterflies, we previously found that genetic variation for plasticity is depleted, both at phenotypic and whole gene expression level, suggesting a limited potential for short-term evolution of plasticity[30]. This is likely due to a history of purifying selection in its highly predictable natural savannah habitat[30].

Here we assess the role of splicing in plasticity, and how it may contribute to adaptation under climate change, by combining transcriptomic and genomic data from the lab and the field. We focus on thorax and abdomen tissues of females, as these tissues and sex are most relevant for studying the life history traits of the seasonal polyphenism. First, we analyse plasticity in the lab and test how variation in splicing, as measured by differential exon expression, is affected by the seasonal environment, genetic background, and their interaction. Second, we complement this with population genomic and molecular evolution analyses of a wild population in Malawi. Specifically, we measure per-gene within-population polymorphism and divergence with an outgroup, testing how these vary across genes that show plasticity-related variation in splicing or expression. Finally, we use event-based splicing analyses to substantiate our exon expression-based approach and link specific splicing event types to variation in population genomic parameters. Our study clarifies the role of splicing for adaptive plasticity, tests its importance compared to gene expression, and reveals its potential to evolve in the wild.

## Results

**Extensive differential exon expression between tissues**. For a broad portrait of transcriptional variation in tissues relevant for the life history differences between seasonal morphs in these butterflies, we compared differences in exon expression between the abdomen (69 samples) and thorax (70 samples) of adult female *Bicyclus anynana* butterflies from seven families in a full-sib design (Supplementary Table 1, 2). Of 15,845 genes in the *B. anynana* genome annotation (v. 1.2), 13,937 had at least two exons, with an average of 8.4 exons per transcript (Supplementary Table 2). A total of 8,533 of these multiexonic genes were sufficiently expressed in our dataset to compare exon expression between tissues. Over half of these genes (50.7%) were differentially spliced between the two tissues. This substantially exceeds estimates of alternative splicing in several other insect species, which range from 4–34% of expressed genes[47]. We attribute this

to a large sample size from a relatively outbred lab population and expect the estimate would increase with even greater spatial and temporal sampling.

These extensive exon expression differences demonstrate that alternative splicing and other sources of isoform variation are common tools for maintaining and accommodating phenotypic variation in these butterflies, in this case, the large functional differences between abdomen and thorax tissues. This conclusion was supported by principal component analysis of normalized exon-level read counts (Supplementary Fig. 1). When counts (Supplementary Fig. 1a) were adjusted to account for average gene-wide expression (Supplementary Fig. 1b), samples clustered strongly by tissue on the first principal components axis, which explained 39.3% of the variance in exon expression among the samples. A heatmap of the top 5,000 differentially expressed exons illustrates significant changes in expression between the two tissues (Supplementary Fig. 1c).

**Smaller role for splicing than expression in seasonal plasticity**. To assess the role of splicing in plasticity and compare it to expression, we investigated patterns of differential exon expression between seasonal morphs with a full-factorial analysis. This design allowed us to dissect seasonal plasticity and genetic differences (using full-sib families as proxy) in alternative splicing, as well as genetic variation in plasticity (the season-by-family

interaction term, SxF), analysing each tissue separately. We also used this analytical framework to investigate differential whole gene expression, permitting direct comparison of plastic and genetic differences between splicing and expression. We have used differential exon expression because the tools allow us to investigate SxF effects, compatible with our study design. However, one potential drawback of focusing on exon expression as a measure for differential splicing is that it may detect differences in exon usage arising from other sources, such as alternative promoters, and may additionally fail to detect some splicing events. While many event-based splicing analyses are unable to test for interaction effects, they provide an important complement to exon expression-based approaches. Thus, we use event-based analyses to corroborate and expand upon our exon expression-based analyses below.

Despite the extensive differential exon expression found between the abdomen and thorax, the effects of seasonal environment on exon usage and expression were remarkably similar between the tissues (Fig. 1a–d; Supplementary Table 2, 3, Supplementary Data 2, 3). As has been found in other phenotypically plastic organisms, an order of magnitude fewer genes were differentially spliced (Fig. 1a) than were differentially expressed (Fig. 1b). Between seasonal morphs, 4.1% of filtered genes (n = 363) in the abdomen, and 2.5% (n = 172) in the thorax, were differentially spliced (Fig. 1a; Supplementary

## a  Differentially spliced genes

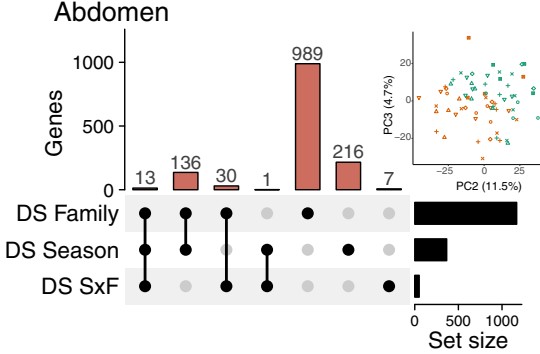
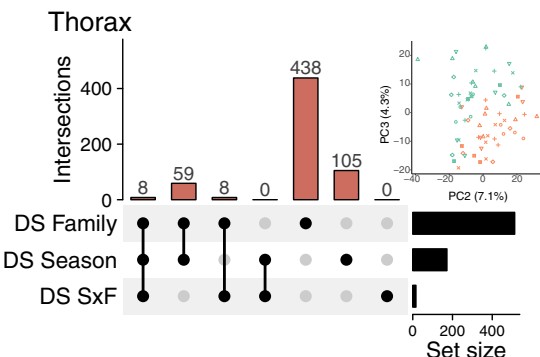

## b  Differentially expressed genes

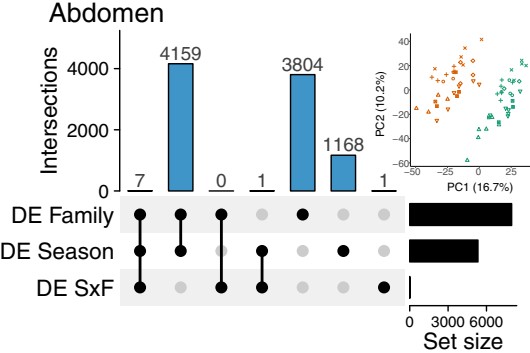
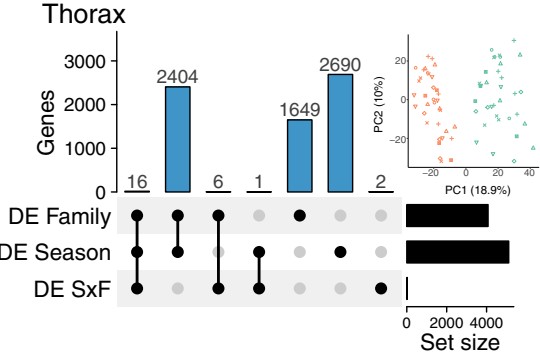

**Fig. 1 Differential splicing and differential expression in two butterfly tissues.** An order of magnitude fewer genes exhibited differential exon expression **a** than exhibited differential whole gene expression **b** in both the abdomen and thorax. There were more genes (set size) with differential exon expression among families than between seasons. Only a small subset of genes showed genetic variation in seasonal plasticity of splicing (SxF). In each upset diagram, dot-plots show the overlap of analyses, vertical bars show the number of genes falling in each category, and horizontal bars show the total number of genes detected as significantly differentially spliced or differentially expressed in each analysis. There was significantly more pairwise overlap between analyses than predicted from the size of the filtered datasets (Supplementary Table 4). Insets show PC plots. Both relative exon expression **a** and gene expression **b** clustered by season (dry = orange, wet = green) and moderately by family (20 = circles, 21 = upward triangles, 30 = plusses, 33 = exes, 53 = diamonds, 60 = downward triangles, 69 = filled squares), although clustering was clearly weaker for exon expression, reflecting the smaller scale of differential splicing compared to differential expression in both tissues. Exon expression was analysed as normalized read counts, adjusted to reflect divergence from average exon expression within the genes. Source data are provided as a Source Data file.

Table 2), while 50.7% (n = 5335) and 57.9% (n = 5111) of multiexonic genes were differentially expressed (Supplementary Table 3). Seasonal differential exon expression effects were substantial enough for morphs to cluster on the second and third PC axes of normalized exon expression within each tissue (Fig. 1a insets). Although patterns were similar, seasonally spliced genes were relatively tissue specific: 36.6% of differentially spliced genes in the thorax were shared with the abdomen, representing only 13.3% of all differentially spliced genes identified across both tissues.

Genotype was a major source of exon expression variation, with over three times as many genes differentially spliced among families as between seasons in both the abdomen and thorax. Focusing on the abdomen results, nearly 85% of family-level variation in exon expression was not associated with seasonal plasticity, either in the form of differential exon expression between seasons or the SxF interaction. In contrast, 41% of among-family differentially expressed genes are differentially expressed for family only, with an additional 46% also involved in seasonal or SxF differential expression, suggesting a larger amount of family-level variation in whole gene expression plasticity. However, we detected very few genes with SxF exon expression (0.6% of multiexon genes) or SxF whole gene expression (0.1% of multiexon genes). These results suggest that splicing is constrained in evolving new plastic responses, but no more so than expression, and lends support for our hypothesis that this outbred laboratory population of *B. anynana* butterflies lacks heritable variation in seasonal splicing plasticity.

**Complementary roles for splicing and expression in plasticity.** Many differentially expressed genes did not have significantly different exon usage, suggesting that exon expression and whole gene expression play nonredundant roles in plasticity. For example, of the genes in the abdomen that were differentially expressed by season (n = 5335), only 4% also expressed different exons by season (Fig. 2a, Supplementary Fig. 2a). Of all the genes differentially spliced by season (n = 366), 40.4% were also differentially expressed by season. In sum, for both seasonal and family effects, many genes were uniquely affected by differential exon expression, although this does represent significantly more overlap than expected by chance (one-sided Fisher's exact test, odds ratio = 1.45, adj. *p*-value = 0.002; Supplementary Table 4). None of the genes exhibiting a significant SxF interaction in differential exon expression were differentially expressed at the whole gene level.

We then assessed quantitatively how differential exon and whole gene expression differ in their regulation of plasticity. Overall, within-gene fold changes in exon expression were smaller than for whole gene expression (Fig. 2b, Supplementary Fig. 2b). Comparing log fold changes at the exon level with those at the gene level for all differentially spliced genes, there was no general relationship between overall levels of gene expression and relative levels of exon expression within those genes (Fig. 2c, Supplementary Fig. 2c). For example, genes with increased whole gene expression in wet morphs did not tend to consistently contain exons with increased expression, as might happen if the seasonal environment consistently increased the retention of skipped exons. Instead, genes with little to no seasonal differences in whole gene expression tended to contain exons with the largest fold changes between seasonal morphs. This lack of correlation between exon and gene expression has been reported in several other studies[17,26]. This may arise from splicing and whole gene expression levels being alternative solutions to selection upon a locus, especially since a large fraction of alternative splicing might affect transcript stability through nonsense mediated decay[48,49].

To evaluate differences in the processes and functions affected by differential exon expression between seasonal morphs and among families, we compared functional annotations (GO terms) of genes that were significantly differentially spliced, expressed, or both. There was very little functional overlap among sets of enriched GO terms (Fig. 2d; Supplementary Fig. 2d, 3–6): GO terms were especially unlikely to be shared between sets of genes with only differential whole gene expression and sets with only differential exon expression. For genes with differential exon expression between seasons in the abdomen, highly enriched terms were related to reproduction, specifically germarium-derived oocyte determination, and transcriptional and cellular component biogenesis regulation (Supplementary Fig. 3, 5, Supplementary Data 4). Terms enriched for differential splicing-only genes in the thorax were related to metabolic processes, regulation of the STAT signalling pathway and germ cell development (Supplementary Fig. 4, 6, Supplementary Data 4). Genes demonstrating both differential exon and whole gene expression between seasons were enriched for GO terms overwhelmingly related to mRNA metabolism in the abdomen and regulation of RNA splicing in the thorax, (Supplementary Fig. 3, 4). In both tissues, sensory perception, especially of temperature, and translation were characteristic of terms enriched for differential whole gene expression-only genes.

**Constraints on seasonally spliced genes in a natural population.** Using individuals from a wild population (Fig. 3a), we quantified pairwise nucleotide diversity ($\pi$) in coding sequences to compare levels among genes with differential exon expression, differential whole-gene expression, or both (Fig. 3b). Previously, we[30] detected elevated $\pi$ and Tajima's D in whole genes that were differentially expressed between seasons when compared with the rest of the genome. These and other results lend support to the hypothesis that plasticity allows for the accumulation of genetic variation within modular regulatory networks[50].

Similarly, we detected increased $\pi$ in whole genes that were differentially expressed between seasonal environments, despite using independent data to estimate nucleotide diversity along with minor differences in our analytical pipeline (e.g., mapping to an annotated reference genome rather than a transcriptome, slightly different filtering parameters, etc.; Fig. 3c, Supplementary Fig. 7a, Supplementary Table 5). However, in genes with differential exon expression between seasons, $\pi$ was consistently lower (8.0% decrease in the abdomen, 10.0% decrease in the thorax), even when these genes were also differentially expressed (6.3% decrease in the abdomen, 9.9% decrease in the thorax). Interestingly, this reduced genetic variation does not appear to be a consequence of alternative splicing itself, but a pattern found exclusively in genes with seasonally plastic splicing; both genes with differential exon expression among families and those with differential whole gene expression among families were associated with statistically meaningful increases in $\pi$ (Fig. 3d, Supplementary Fig. 7c, Supplementary Table 5). These findings were also supported by direct comparisons between season and family groups in both tissues, as $\pi$ was statistically higher in genes that had differential exon expression among families than those that had differential exon expression between seasons, while genes with differential exon expression in both analyses (i.e., between seasons among families) had intermediate $\pi$ values (Supplementary Fig. 8). Too few genes had an SxF interaction in either exon expression or whole gene expression to allow for statistically robust comparisons (Supplementary Fig. 7).

We hypothesize that these differences in $\pi$ between differentially expressed and spliced genes arise from an interaction between selection on the seasonal polyphenism and our ability to

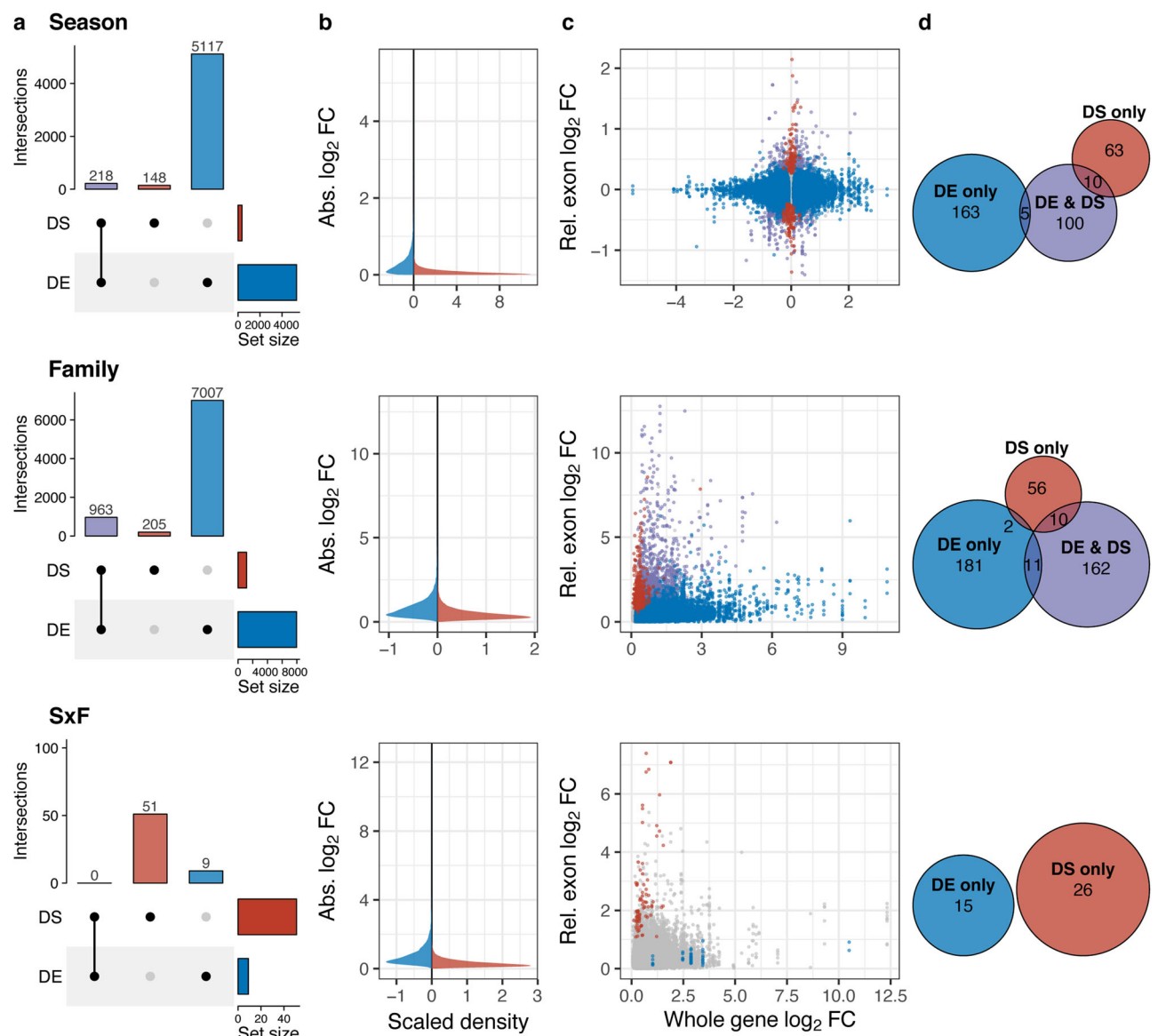

**Fig. 2 Alternative splicing plays a smaller but complementary role to whole gene expression in seasonal plasticity.** Results presented are for the abdomen, similar patterns were found in the thorax samples (Supplementary Fig. 2–6). **a** There was significantly more pairwise overlap (purple) of differentially spliced (DS, differential exon expression; red) and differentially expressed (DE, differential whole gene expression; blue) gene sets for the main effects of season, (one-sided Fisher's exact test, OR = 1.45, $P = 1.5 \times 10^{-2}$, Supplementary Table 4) and family (one-sided Fisher's exact test, OR = 1.57, $P = 1.70 \times 10^{-7}$) than predicted from the size of the filtered datasets, while genes with a season-by-family interaction (SxF) in splicing or expression did not overlap at all. **b** Effect sizes of differential splicing (e.g., relative exon log fold changes, red) were generally smaller than those of differential expression (e.g., whole gene log fold change, blue). For seasonal comparisons, fold change represents the change in expression from dry to wet. For family comparisons and SxF interaction, we used the maximum absolute fold change among all families as a proxy for fold change. **c** There was no relationship between whole gene log fold change and exon log fold changes when compared between season, among families or SxF. **d** Euler diagrams show that very few gene ontology (GO) terms enriched for genes that are differentially expressed (blue), both differentially expressed and differentially spliced (purple), or only differentially spliced (red) overlapped between these gene sets. Circle sizes correspond to the total number of enriched GO terms (two-sided Fisher's Exact Tests, parentChild algorithm, p-value < 0.05), and in cases where the number of shared terms is very small, the number has been placed adjacent to the intersection. Source data are provided as a Source Data file.

detect these effects. Genes directly affecting the seasonal polyphenism are likely historical targets for positive selection, as well as ongoing purifying selection, and therefore we expect them to have lower levels of genetic variation on average than other loci. However, as in any group of differentially expressed genes, we expect such causal genes to comprise only a small fraction of all the seasonally differentially expressed genes that we observed. Causal genes are, in turn, expected to initiate cascades of diverse expression networks. Without any ability to

discriminate which among these thousands of genes are under strong selection, seasonally differentially expressed genes as a group are not expected to have lower π. Rather, stabilizing mechanisms that buffer alternative phenotypes against genetic variation are predicted to allow for the accumulation of polymorphism in genes that are not subject to direct purifying selection[1,6,51]. Furthermore, conditionally expressed genes may experience relaxed selection because they are only expressed in certain environments[52,53]. In contrast, differentially spliced genes

## Nucleotide diversity

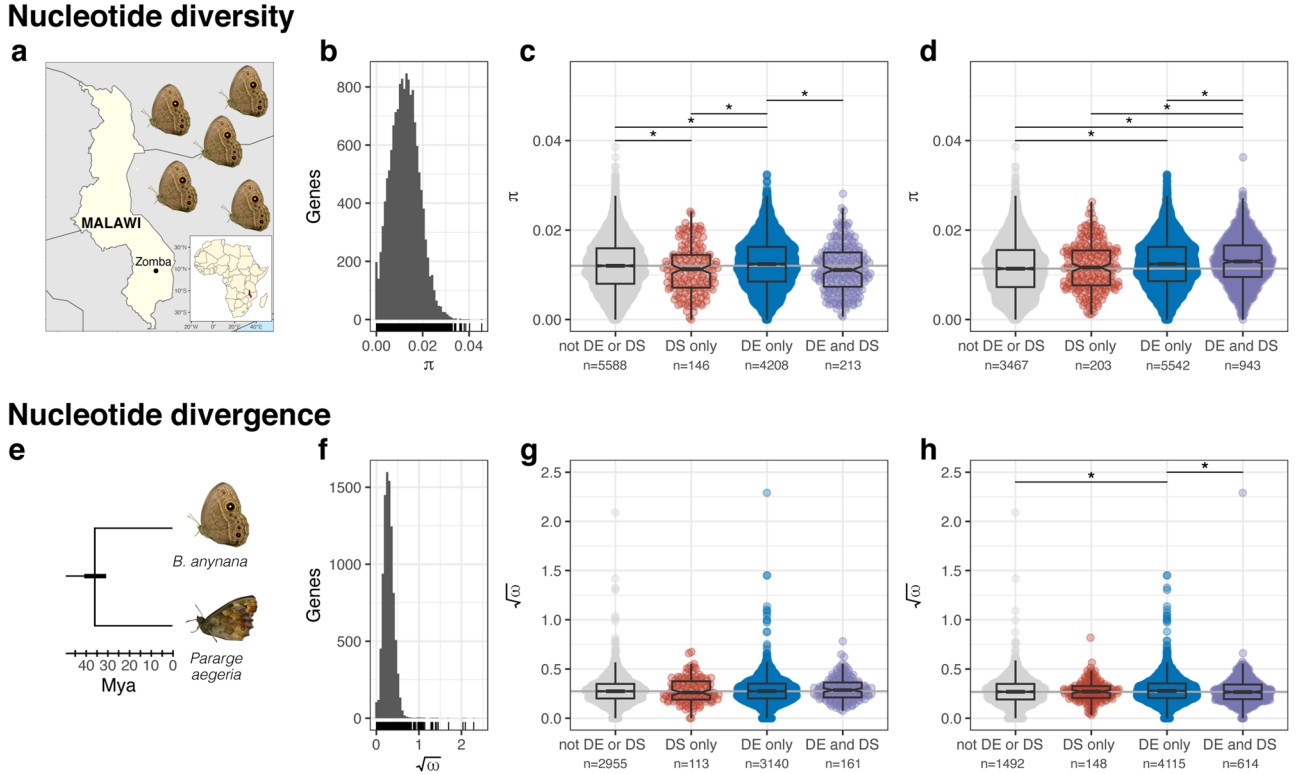

## Nucleotide divergence

**Fig. 3 Short term but not long-term constraint depends on molecular mechanisms of plasticity. a** Five butterflies were collected from Zomba, Malawi, and used for population genomics. **b** Nucleotide diversity (π) was calculated for 14,402 genes in the *B. anynana* annotation, 10155 of which were multiexon. A histogram represents the distribution of π values, while a rug plot show values for individual genes. **c** Seasonally differentially spliced (DS, differential exon expression) genes had reduced π, whether the genes were also differentially expressed (DS only = red, DE and DS = purple). **d** In contrast, genes that were DS or DE among families tended to have increased π relative to the rest of the genome. **e** Nucleotide divergence (ω) was calculated between *B. anynana* and *Pararge aegeria*, which shared a common ancestor 31–41 mya[49]. **f** ω was calculated for 9306 single-copy orthologs, 6369 of which were multiexon, and square-root-transformed for analyses. **g** There was no difference in ω among genes that were differentially expressed or differentially spliced by season. **h** Among families, DE-only genes showed higher ω, although this pattern was inconsistent across tissues. **c, d, g, h** Data shown are based on abdomen samples (see Supplementary Fig. 7, 9 and Supplementary Table 5, 6 for figures and analyses based on thorax samples). Asterisks (*) denote statistically meaningful differences (Bayesian linear models, Supplementary Fig. 7A, C and Supplementary Fig. 9A, C). Point clouds represent π or $\sqrt{\omega}$ values for each gene set. Values are summarized with boxplots: the center line represents the median, the box encloses the 25th-75th quartiles and is notched (median +/− 1.58 * interquartile/$\sqrt{n}$), and whiskers extend to 1.5x the interquartile range. The grey horizontal line indicates the median of genes that were not DE or DS. Source data are provided as a Source Data file. Maps were constructed in R[100,101] and modified for publication with Inkscape[102]. *B. anynana* photo was modified with permission from O. Brattström. *P. aegeria* photo was modified with permission from O. Lindestad. Phylogeny schematic was made by the authors from data published in Pena et al.[56].

have been reported to have upwards of 50% of their splicing variation due to *cis*-regulatory variation[33,34]. Mis-regulation of splicing within alternative morphs is presumably costly, even when selection is episodic. Thus, though the overall number of seasonal differentially spliced genes is much lower than differentially expressed genes, most loci in the differentially spliced group are expected to have a lower π due to selection upon the seasonal polyphenism, which is what we observe. In contrast, both types of transcriptional variation are expected to have slightly higher π in the among-family comparison because the family-level grouping detects patterns of heritable variation, though this is expected to be neutral.

Next, we investigated whether this strong selection for the seasonal polyphenism may leave long-term signatures of selection in the genome of *B. anynana*. While alternatively spliced exons have been shown to diverge more rapidly than constitutively spliced exons in mammals[54,55], a small body of evidence suggests that genes that are differentially spliced between alternative sexes tend to be functionally and evolutionarily constrained, with lower values of nucleotide divergence[27]. To assess divergence, we calculated the ratio of nonsynonymous to synonymous nucleotide

divergence (ω) between single copy orthologs shared between the published genomes of *B. anynana* and *Pararge aegeria* (Elymniini, Parargina; Fig. 3e). The subtribes Mycalesina and Parargina diverged 32–41 Mya[56], and while both retain seasonally plastic phenotypes including forms of reproductive or developmental diapause, this phenotype has certainly been lost, gained and altered over evolutionary time[57]. We observed no consistent evidence in either tissue for an effect of seasonal differential exon expression on nucleotide divergence (Fig. 3g; Supplementary Fig. 9, Supplementary Table 6). The only statistical support was for elevated nucleotide divergence in differentially expressed genes among seasons in the thorax and families in the abdomen (Fig. 3h, Supplementary Fig. 9, Supplementary Table 6). Thus, unlike the observed patterns in π which were consistent across analyses and tissues, we were unable to detect long-term selection patterns associated with either differential exon expression or differential whole gene expression.

**Nucleotide diversity differs among specific splice events**. To test the hypothesis that different splice event types are contributing differently to the erosion of genetic variation observed in

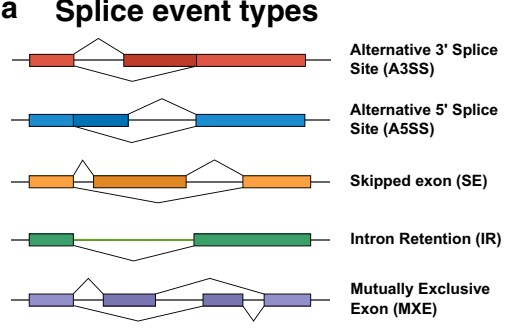

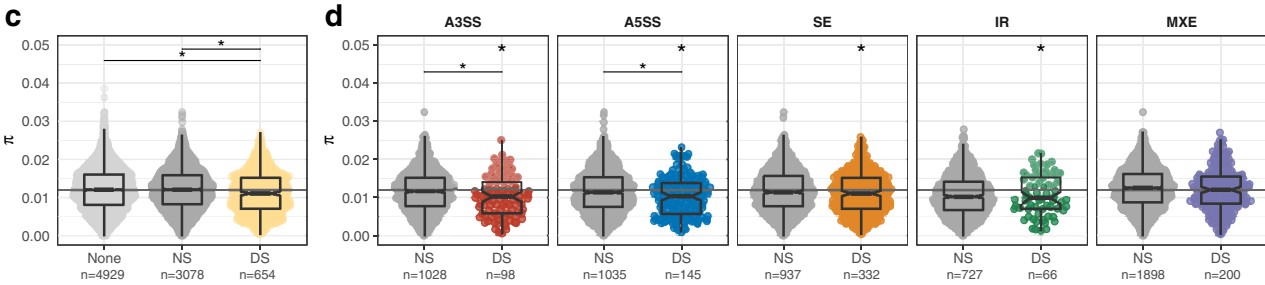

**Fig. 4 Erosion of genetic variation differs among alternative splice event types. a** We analysed five major splice event types: alternative 3′ (A3SS) and 5′ splice sites (A5SS), skipped exons (SE), intron retention (IR), and mutually exclusive exons (MXE). In the abdomen, **b** small subsets of detected splice events were significantly differentially spliced (coloured bars) between seasonal morphs. The panel to the right zooms out to show all detected events (dark grey bars outlined in black). **c** Nucleotide diversity ($\pi$) among genes with no splicing events (None, light grey), nonsignificant splicing (NS, dark grey) and differential splicing by season (DS, yellow) indicate that seasonally spliced genes have decreased polymorphism. The horizontal grey line indicates the median $\pi$ for genes with no splicing ("None"). **d** For genes containing one event type, $\pi$ was lower in DS genes compared to both NS genes across all event types. The grey line extends from **c**. DS genes for all event types were also different from genes with no splice events **c, d**. Point clouds represent $\pi$ values for each gene set. Values are summarized with boxplots: the center line represents the median, the box encloses the 25th–75th quartiles and is notched (median ± 1.58 * interquartile/$\sqrt{n}$), and whiskers extend to 1.5x the interquartile range. An asterisk (*) centred over a horizontal line between groups indicates a meaningful statistical difference between these groups whereas an asterisk centred over the DS points indicates a statistical difference from the grey line (Bayesian linear models, Supplementary Fig. 11A, 13A). See supplementary Fig. 10, 11, 13 and Supplementary Table 8, 9 for figures and analyses based on thorax samples. Source data are provided as a Source Data file.

seasonally spliced genes, we quantified splice events between morphs, assessed the maintenance of functional isoforms, and compared the effect of differential splicing on nucleotide diversity for each event type. The various splice event types (Fig. 4a) that contribute to alternative splicing differ both in how they are regulated (*cis vs. trans*) and in their potential to result in nonsense mediated decay of non-functional mRNAs versus functional protein isoforms[49,58]. Based on our previous results, we expected genes containing events that are differentially spliced between seasonal morphs to experience stronger selection and to have lower genetic variation. Further, we expected that this signal would be strongest for splice events that are more likely to be *cis*-regulated, including alternative 3′ and 5′ splice sites (A3SS, A5SS) and intron retention (IR).

Event-based analyses identified all alternatively spliced events (Fig. 4b; Supplementary Fig. 10, Supplementary Table 7, Supplementary Data 5) and the subset of events that were significantly differentially spliced between seasonal morphs (Fig. 4b; Supplementary Fig. 10b, zoom panels). In both the abdomen and thorax, skipped exons (SE) and mutually exclusive exons (MXE) were most abundant, while IR events were least abundant. Between 6.5% and 17.5% of detected events were differentially spliced. There were more differentially spliced SE events than expected, whereas the remaining splice types had

fewer significant events than expected (Fisher's exact test, df = 1, adj. *p*-value = 6.60E-20; Supplementary Table 7). Skipped exon and MXE events were most likely to retain functional open reading frames (ORFs), regardless of whether these events were significantly different between seasonal morphs. Both alternative A3SS and A5SS, which involve changes at the ends of otherwise constitutively expressed exons, were more likely to retain putative ORFs when differentially spliced between seasonal morphs than when this splicing was not significantly different. We note, however, that our analysis of ORFs disregards additional sources of nonsense mutations, such as premature stop codons, and therefore provides a conservative estimate of nonsense mediated decay. Nevertheless, it suggests that A3SS and A5SS splice event types that are critical to seasonal plasticity may be under selection to maintain functional variants.

We found additional support for evolutionary constraint in differentially spliced genes, defined here as genes containing at least one event significantly differentially spliced between seasons. This category had lower $\pi$ than both genes containing alternatively spliced events and multiexonic genes without splicing (Fig. 4c, Supplementary Fig. 11a, b). When compared with differentially expressed genes from the previous analyses, we once again found distinctly lower nucleotide divergence in differentially spliced genes in both the abdomen and the thorax

(Supplementary Fig. 12a, b). By event type, only genes containing differentially spliced MXE events did not show a similar pattern of genetic erosion (Fig. 4d, Supplementary Fig. 13a, b, Supplementary Table 8). Both SE and MXE events involve cassette exons that are included or excluded to produce alternative protein products, and generally depend on more complex regulatory mechanisms that other splice event types. For example, greater trans-regulation was found for SE compared to ASS and IR in *Drosophila melanogaster*[33]. Accordingly, we found differentially spliced A3SS and A5SS events had the greatest decrease in nucleotide diversity. As A3SS and A5SS events are generally more likely to be *cis*-regulated or associated with *cis*-regulatory elements[59], their decreased $\pi$ is consistent with expectations of positive and purifying selection acting upon these loci.

## Discussion

Differential whole gene expression has long been recognized as crucial to both the development and maintenance of plasticity[60–62]. Our results add to the mounting evidence that differential splicing affects a smaller but nonredundant subset of genes compared to differential expression. Furthermore, we found that the greatest seasonal differences in splicing tended to fall in genes with smaller differences in expression. Differential splicing is therefore introducing protein variation for genes that remains uncaptured in analyses of average whole gene expression, and therefore represents an important route to identifying and studying genes involved in phenotypic plasticity.

The evolution of seasonal splicing appears to be highly constrained. We recovered very few genes with a season-by-family interaction for splicing, suggesting a lack of standing genetic variation for seasonal splicing plasticity. We also show strong evidence for reduced nucleotide diversity in seasonally differentially spliced genes, using both exon expression and event-based approaches. Growing evidence suggests a history of purifying selection in highly predictable fluctuating environments results in a loss of genetic variation[30,63]. This hypothesis has been difficult to test in the context of differential whole gene expression since the vast majority of differentially expressed genes are unlikely to be gene expression variation is expected to be generated by *trans*-acting splicing factors. Thus, the complexity of regulatory networks can mask the genes underlying this plasticity, consequently undermining tests for standing genetic variation for seasonal plasticity. Some approaches, like allele-specific expression, provide a means of identifying genes involved in this variation[64,65]. By focusing on differently spliced genes, we have uncovered patterns of reduced genetic diversity associated with seasonal plasticity, consistent with them being targets of purifying selection. Our findings suggest that differential splicing may provide an alternative route to detecting important genes underlying plastic phenotypes, potentially providing access to components of regulatory cascades involved in compensatory responses to alternative seasonal morphs.

## Methods

**Data acquisition, study organism and experimental design**. We used RNA sequence data from abdomen and thorax samples of female *Bicyclus anynana* butterflies available from the National Centre for Biotechnology Information (NCBI) Sequence Read Archive (SRA) BioProject ID PRJNA376691. These sequence data were previously used to investigate gene expression differences between dry and wet adult seasonal morphs[30]. The butterflies were from a captive, outbred laboratory population. Larvae from seven families were reared in a split-brood design, with females from seven families reared at 19 °C (dry season conditions) and 27 °C (wet season conditions; Supplementary Table 1). Larvae also experienced two food stress conditions: fifth instars fed either ad libitum on maize leaves (control) or on nutrient-free agar (stress; to avoid dehydration). Details of butterfly handling, body part sampling, sequencing and initial quality filtering can be found in Oostra et al.[30]. Briefly, purification, library preparation and sequencing

(paired-end, 2 × 100 bp, mean insert size 350 bp, Illumina HiSeq 2000) of isolated RNA was performed by BGI (People's republic of China). Raw reads were trimmed using bbduk2 (trimq = 20, bbmap v. 35.69), with an average of 6.24% (95% CI: 5.92–6.56%) of reads trimmed per sample (Supplementary Data 1).

**Read mapping**. We mapped reads to the *B. anynana* genome (v. 1.2; GCA_900239965.1) and associated annotation (GFF) available on NCBI BioProject PRJNA434100 using STAR (v2.5.0[66]), a highly sensitive splice-aware RNA-sequence aligner that identifies splice junctions with high precision[67]. We generated the genome directory and mapped reads using default settings and a two-pass approach. Ninety-three percent of reads mapped successfully (95% confidence interval: 92.4–93.1%; Supplementary Data 1). On average, STAR identified $6.39 \times 10^6$ (95% CI: 6.28–6.50 × 10^6) splice junctions among the trimmed reads in each sample. Of these junctions, most were based on the reference annotation while very few (95% CI: 0.249–0.258%) were novel. Reads were sorted and indexed with Samtools (v.1.9). For RNA-sequencing methods that use poly(A) selection, RNA degradation has the potential to bias coverage to the 3′ end of the transcript[68], which might influence the ability to detect both differential exon expression and splicing events. In order to exclude such potential bias, we evaluated normalized coverage across the length of annotated transcripts using Picard tools (v. 1.139)[69] and found no evidence for RNA degradation in either the abdomen or thorax samples (Supplementary Fig. 14). Rather, we detected a slight 5′ bias that is characteristic of intact RNA samples[68].

**Differential exon expression**. Mapped reads were quantified at the feature level using *featureCounts* (GTF.featureType = exon), which we ran through Rsubread (v2.0.0[70,71]) in the R statistical environment (v. 4.0.0). *featureCounts* successfully assigned an average of 79.3% (95% CI: 78.8–79.9%) of aligned reads. We filtered low-expression exons using the *filterByExpr* tool in the edgeR package[72]. For seasonal and family comparisons, we analysed the abdomen ($n = 69$) and thorax ($n = 70$) samples separately and applied filters across all samples within each seasonal morph group. This conservative filtering allowed us to detect season-limited exon expression, but limited detection of exons that were only expressed in one family or season-by-family interaction group. Reads were successfully assigned to 118,950 exons. For the between-tissue comparison, the filtered data retained 64,832 exons. Within tissues, our filtered data retained 68,978 exons in the abdomen samples and 46,289 in the thorax (Supplementary Table 2). Filtered and normalized (*calcNormFactors*, method = TMM) exon counts were visualized using principal component analysis (*prcomp*, stats package[73]). Exon expression counts output by featureCounts are not corrected for whole gene expression, meaning that differences in exon counts between samples could be caused by differential exon expression, differential whole gene expression or both. By calculating the residual of the normalized (TMM) exon expression of all exons within each gene, we can compare exon expression across samples without the confounding effect of differential whole gene expression. We calculated this "TMM-residual expression" by subtracting exon TMM values from the average TMM across all exons within a gene within an individual. These values were compared using PCA.

Differentially expressed exons were analysed in edgeR (*glmQLFit*, *glmQLFTest*). For the between-tissue comparison, the glmQLFit object was fit with tissue as the predictor. Within tissue, the glmQLFit object was fit on a design matrix with the fully factorial comparison of season by family (~0+ season: family). As multidimensional scaling of gene and exon read counts did not reveal clustering by diet on any of the first five axes, we did not test the effect of food stress in splicing analyses. Normalized exon expression was compared within genes to identify exons with log-fold-changes that differed from the within-gene average (*diffSpliceDGE*). *diffSpliceDGE* can perform contrasts between two groups. Thus, the effect of tissue and the effect of season on splicing was tested using single pairwise contrasts: all abdomen against all thorax samples, and within tissue, all dry-season against all wet-season individuals. However, to test the main effect of family and the interaction effect of season and family, we tested all 21 relevant pairwise contrasts (e.g., family 21 against family 29 individuals, etc.). For all contrasts, we used the Simes method to calculate *P*-values for differential splicing at the gene-level, corrected for multiple testing within the edgeR pipeline using a Benjamini-Hochberg correction (FDR). For the season analysis, we accepted adjusted *P*-values below 0.05, which produced a list of significantly differentially spliced (DS) genes. For the family and SxF analyses, we accepted genes as differentially spliced if at least one contrast had an adjusted *P*-value below 0.05. Exon fold changes, relative to other exons within the same gene, were calculated directly in the season analysis. The family and SxF interaction analysis produced a log fold change estimate for each contrast, so to summarize the effect-size for each exon we calculated the range of log-fold changes across all contrasts.

**Differential gene expression**. Gene expression was quantified using *feature-Counts* (useMetaFeatures geneid) and analysed in edgeR following a similar pipeline as differential gene expression. Of the 15,845 genes in the published annotation, our filtered datasets retained 11,364 and 9,516 genes in the abdomen and thorax samples respectively. Differential gene expression was analysed using an ANOVA-like framework, testing the main and interaction effects of season and family on normalized counts of the filtered reads, with the following model

formulas: counts ~ season + family + season: family and counts ~ season + family. By specifying the interaction contrasts, the first model produced estimates for the season-by-family interaction (SxF) term in an ANOVA-like framework. The second model estimated the main effects of season and family. To focus on differential expression of genes that could also be differentially spliced, we excluded all genes with only one exon ($n_{ab}$ = 841 genes, $n_{th}$ = 681 genes, Supplementary Table 2).

The overlap of differentially spliced and differentially expressed gene sets were compared using Fisher's exact tests implemented in R (*GeneOverlap* package v.1.24.0[74]) adjusted (BH method) for multiple testing, and visualized using Upset diagrams (*ComplexHeatmap* package, v.2.4.3[75]). We further compared the overlap of splicing and expression by plotting the relative log fold changes of exons (described above) to the log fold changes of differentially expressed genes. As with exon expression, the analysis of whole gene expression for family and the SxF interaction produced multiple logFC estimates for each gene, which were summarized by calculating the range across all families.

**Gene set enrichment**. We generated a new gene ontology annotation of the *B. anynana* genome using eggNOG (v.50[76]). The GO annotation included 12,314 genes that were used by the R package topGo (v.2.28[77]) to calculate enrichment of three gene sets (genes that were DE only, those that were differentially spliced only [exon expression-based], and those that were both differentially expressed and spliced) for each of the three analyses (between seasons, among families, season-by-family) in the abdomen and thorax, for a total of 16 gene sets. We visualized the functional overlap among gene sets in two ways. We displayed the overall overlap in enriched functional terms—as determined by topGO—between gene sets with Euler diagrams (eulerr package, v.6.1.0[78]). Second, we ran enriched GOterms for each gene set (i.e., differentially spliced-only differentially expressed only, differentially expressed and spliced) through REVIGO[79] to cluster terms and identify similarity and dispensability. These sets were further arranged into clusters of semantic similarity, or the degree of similarity between the GO term descriptions in the Gene Ontology (go.obo) and UniProt-to-GO databases, with the package simplifyEnrichment[80,81]. The overlap in functional terms between DE, DS and DE-DS gene sets was visualized as a heat map of enrichment scores [$-\log_{10}$ (p-value)].

**Nucleotide diversity of a wild B. anynana population**. We generated whole-genome DNA resequencing data from a wild *B. anynana* population from Zomba, Malawi (15°22′S, 35°19′E). This is ca. 430 km from the location of the original Leiden laboratory population established in 1988[41]. The Zomba population was collected and brought to the laboratory in March 2007 and has been studied previously in phylogeographic analyses using candidate genes[82,83]. Butterflies were frozen alive at −80 °C and stored at that temperature, excepting short periods of freezer malfunction.

We selected five females for whole-genome sequencing. We used ca. 5 mm tissue cut from the proximal part of the abdomen, avoiding eggs. We extracted genomic DNA using Qiagen's DNeasy Blood & Tissue kit (Qiagen, cat. no. 69504) following manufacturer's protocol. We disrupted tissues in buffer ATL using a Qiagen tissue lyser (50 Hz for 2 × 3 min) and incubated in Proteinase K for 3.5 h at room temperature. We then included a 2-min room temperature RNase incubation (Qiagen RNase A 100 mg/ml) prior to column purification. After elution, DNA quantity was confirmed using Qubit and Nanodrop. Next, DNA was bead purified and quality was assessed on a fragment analyzer. Illumina fragment libraries were prepared by the Centre for Genomic Research (University of Liverpool) using the NEBNext Ultra II FS kit (New England BioLabs, cat. no. E7805) on the Mosquito platform with 1/10 volumes (SPT Labtech) and quantified using qPCR. Libraries were sequenced over ¼ of a lane on the Illumina NovaSeq using S4 chemistry, with 2 × 150 bp paired-end reads.

Raw sequencing reads were quality-assessed using FastQC (v. 0.11.4) and trimmed using Cutadapt (v. 1.2.1[84]) and Sickle (v. 1.2[85]) using a minimum window quality score of 20 and discarding reads shorter than 15 bp. Sequencing depth after trimming ranged 53–96 million reads (mean: 77 M). We then mapped trimmed reads to the *B. anynana* v1.2 reference genome using bwa-mem (v. 0.7.15-r1140[86]) with standard settings and removed duplicated reads using Picard tools (v. 2.0.1[69]). Coverage ranged 25–46× (mean: 37×). Next, we used angsd (v. 0.931-10-g09a0fc5[87]) to compute polymorphism from the aligned reads. First, we computed a global maximum likelihood estimate of the folded site frequency spectrum based on the whole genome, using the reference genome as both reference and ancestral state, and removing bad alignments (minimum read quality > 12 and minimum mapping quality > 19). Next, again in angsd, we estimated per-site polymorphism (nucleotide diversity π) using the global spectrum as a prior. Finally, we used the *B. anynana* reference annotation[88] to calculate (in R v. 3.6.3) polymorphism per exon, and to average across each whole gene, weighting the per-exon estimate by its length. This resulted in estimates of nucleotide diversity π for coding sequence of each gene.

**Divergence from an outgroup**. Single copy orthologs (SCOs) between *B. anynana* and the nymphalid butterfly *Pararge aegeria* (available from NCBI, PRJEB28004, https://www.ncbi.nlm.nih.gov/assembly/GCA_900499025.1/) were identified with OrthoVenn2[89], (https://orthovenn2.bioinfotoolkits.net; last accessed 01/2021). The longest isoforms of SCOs were identified for each species and CDS were aligned,

codon aware, with MACSE (v.1.01[90]). Nucleotide divergence (ω) was calculated using HyPhy (v.2.5.25[91]) between 9306 aligned SCOs, out of 9362 identified by OrthoVenn2; two were excluded as outliers (ω = 42.08 and 37.79).

**Statistical analyses of nucleotide diversity and divergence**. Nucleotide diversity (π) was compared between gene sets using Bayesian linear models (brms package v. 2.14.4[92,93]). Models took the form, π ~ predictor, where π was nucleotide diversity and the predictor was a categorical variable identifying gene sets (e.g., differentially spliced-only differentially expressed only, differentially expressed and spliced, etc.). Models had gaussian distributions and uninformative priors for both the intercept and all predictors. Three chains were run with 500 warmup iterations followed by 4500 sampling iterations. Support for differences in means and variances were estimated by comparing model posteriors (*hypothesis* function, brms package). These analyses were supported with ANOVAs and pairwise multiple comparisons (Tukey's HSD, rstatix package[94]).

Similarly, nucleotide divergence (ω) was compared using Bayesian distributional models, which not only allowed us to compare unequal means but also account for unequal variances between groups. To account for strong right skew, nucleotide divergence (ω) was square-root transformed and modelled using a skew-normal distribution. Models took the form, (ω ~ predictor, σ ~ predictor), where ω was nucleotide divergence, σ was the variance, and the predictor was a categorical variable identifying gene sets. Again, model posteriors were compared using the brms hypothesis testing framework. Analyses were checked using nonparametric Wilcoxon or Kruskal-Wallis rank sum tests (rstatix package[94]).

**Splice events**. We used the event-based tool rMATS (turbo v.4.1.0[95]) to quantify both annotated and novel splice junctions in the abdomen (n = 69) and thorax (n = 70), and to assess differential splicing between the two seasons. The rMATS program identified five types of splice event: alternative 3′ splice site (A3SS), alternative 5′ splice site (A5SS), skipped exon (SE; also called cassette exons), intron retention (IR), and mutually exclusive exons (MXE), by default including all events with at least one read supporting the exon inclusion form and the exon skipping form in either the abdomen or thorax samples. Raising this threshold caused the number of splice events to drop steeply, as did adjusting the minimum number of samples with at least one read supporting the inclusion or exclusion form. Based on a comparison of read thresholds, we used an in-house script to filter out events that lacked support (at least five reads for both the inclusion and exclusion form) in three individuals or fewer in each tissue. To be considered a significantly differentially spliced event, we required splice sites to have adjusted p-values below 0.05 and a ΔPSI value of >5%, a cut-off that was chosen based on other event-based differential splicing analyses[95–97]. Following Grantham and Brisson[17] we tested whether significant splice events were over or underrepresented in each splice type category using Fisher's exact tests (rstatix package). The expected number of differentially spliced sites was calculated as the product of the subtotal of each splice type and the proportion of total splice events that were significant (splice type subtotal * [total significant splice events/total expressed splice events]). Results were corrected for multiple comparisons using the BH method.

We evaluated whether alternative splicing was likely to produce non-functional transcripts that would be targets of nonsense mediated decay. rMATS event-based approach does not allow for direct comparisons of isoforms, so we approximated open reading frames by checking whether retained introns or skipped, mutually exclusive or alternative exons were multiples of three, thereby maintaining the reading frame in relation to the primary transcript.

In many cases, the splice event analysis identified multiple events per gene. To compare nucleotide diversity and divergence, we identified genes as differentially spliced (DS) when they contained at least one event meeting our significance thresholds. Genes that containing splicing events that were not significant were designated as simply alternatively spliced (NS). We compared all these gene sets to the remaining multiexonic genes in which no alternative splice events were detected ("None", grey) using the statistical analyses described above (Statistical analyses of nucleotide diversity). To corroborate exon-based analyses, we further compared the nucleotide diversity and divergence of these event-based DS genes to genes that were differentially expressed between seasons using Bayesian linear and distributional models. These and all other analyses and visualizations performed in R were supported by the tidyverse[98] and ggpubr[99] packages. The map of Malawi featured in Fig. 3a was made using the rgeos[100] and afrilearndata[101] packages. All figures were modified for publication with Inkscape[102].

**Reporting summary**. Further information on research design is available in the Nature Research Reporting Summary linked to this article.

## Data availability

*Bicyclus anynana* RNA-seq data used to estimate differential splicing and differential expression in this study were accessed from NCBI archives (PRJNA376691). We accessed the *B. anynana* genome v1.2 from NCBI, PRJNA434100, (https://www.ncbi.nlm.nih.gov/genome/10970?genome_assembly_id=358767). Illumina short-read whole genome data generated in this study and used to estimate population genetic parameters were archived at NCBI under accession number PRJNA786886. The genome of *Pararge aegeria* that was used to identify single copy orthologs and calculate nucleotide divergence was

accessed from NCBI, PRJEB28004, https://www.ncbi.nlm.nih.gov/assembly/GCA_900499025.1/). Metadata and results of differential expression and spicing analyses have been included as Supplementary Data files. Source data necessary to perform subsequent analyses are provided in the SourceData_B_anynana_AS.zip file, as described in Supplementary Note 1 in the Supplementary Information file. Source data are provided with this paper.

## Code availability

Bash and R scripts for generating lists of differentially spliced and differentially expressed genes are provided at https://github.com/rstewa03/B_anynana_differentialSplicing[103]. Code necessary to perform subsequent analyses is available as Source Data (SourceData_B_anynana_AS.zip).

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

## Acknowledgements
We thank three anonymous reviewers for their invaluable feedback. We also thank H. Dort, K. Tunström and B. Willink for suggestions and helpful discussions. This research was supported by Carl Tryggers Stiftelse anslag (CTS 18:415 to C.W.W. and R.A.S.) and the Swedish Research Council (2017-04386 to C.W.W.). Fieldwork was supported by the Earth and Life Sciences programme of the Netherlands Organization for Scientific Research (Grant no. 814.01.012) to M.A.dJ. DNA and RNA sequencing was supported by the University of Liverpool (Technology Directorate Voucher Scheme grant 122654 / SD2994_R2 to V.O.), and by the European Union (Marie Skłodowska-Curie Fellowship 660172 to V.O. and Network of Excellence LifeSpan FP6 036894, and IDEAL FP7/2007-2011/259679), respectively. Population genomic analyses were conducted with the aid of computational resources (Puhti supercomputer) provided by CSC – IT Center for Science, Finland. We thank the University of Liverpool's Centre for Genomic Research for bioinformatic support.

## Author contributions
The study was conceived by V.O. and C.W.W. Wild butterflies were collected by M.A.d.J., population genomic parameters were calculated by V.O. and C.W.W., while R.A.S. analysed the transcriptome and presented all the data. The ideas and text of the manuscript were developed and written by R.A.S., V.O. and C.W.W. All authors contributed to subsequent revisions.

## Funding

## Competing interests
The authors declare no competing interests.
