## [Peer Review File · Nature Communications]

Reviewers' Comments:

Reviewer #1:

Remarks to the Author:

Phenotypic plasticity is a well known phenomenon where animals produce alternate phenotypes in response to changing environmental conditions using a single genotype. In this manuscript Steward et al. investigate the role of alternative splicing in seasonal plasticity in *Bicyclus anynana*, a textbook model of seasonal polyphenism. While previous reports in this system have investigated some physiological mechanisms underlying plastic wing pattern elements (Monteiro et al. 2015, Bhardwaj et al. 2018), little is known about alternative splicing modulating such plasticity. Steward et al. have analysed effects of seasonality and genetic background on plasticity in a lab population, and further expanded the study to include comparative study with population genomics study of a wild population.

Steward et al. found signatures of differential splicing across tissues, while few genes had seasonal splicing or expression. It was interesting to note that differentially expressed genes were not differentially spliced, which suggests both have a complementary non-redundant role to play in regulation of plasticity. Further they discuss constraints on evolution of plasticity in the natural population and lack of standing genetic variation in existing populations, as well as both cis-regulatory and trans environments as interesting candidates for future studies.

The work is excellent, experimental design neat and conclusions clear and concise. Supplementary materials were easy to follow through. The results are of significance to anyone working in the area of alternative splicing and phenotypic plasticity. I thoroughly enjoyed reading the manuscript and recommend it for publication.

Suggestions :

Why were abdomen and thorax chosen as test samples to study plasticity and alternative splicing, when wings which show the most striking seasonal differences in the developmental life history of butterflies, excluded from the study? I believe that has to do with data availability at NCBI, but a discussion on relevance would be appreciated.

Did you find any sex-specific patterns of alternative splicing in these butterflies? It might additionally be worth discussing patterns of plasticity which is sex-limited and role reversal in alternate seasons (in context of Prudic et al. 2011).

It might be out of the scope of this study, but I would be curious to know if any of the genes you found in this study overlapped with genes such as trehalase, cortex or herfst, as found in van der Berg et al., 2020.

Minor Edits :

Line 350 : Please change to nonsense mediated decay.

Line 602 : Grantham and Brisson (check comma)

Reviewer #2:

None

Reviewer #3:

Remarks to the Author:

The authors investigate the role of alternative splicing in seasonal plasticity. They present a small subset of genes that are alternatively spliced between seasons, especially when family is taken into account. Testing for overlap between differentially expressed genes and differentially spliced genes suggested splicing as an alternative to differential gene expression. The authors show

reduced genetic variation in genes with alternative splicing events.

Overall, I think the study of alternative splicing in seasonal plasticity is interesting. Investigating evidence of selection in genes associated with seasonal plasticity is especially important, as that topic has been relatively understudied. I have some major concerns with the methodology and the presentation, that I think the authors should address.

1. The selection of alternatively spliced genes. If I understand correctly, alternative splicing was determined by differential exon expression within genes. This makes the assumption that genes were evenly sequenced across the transcript, an assumption that is not addressed at all. The authors could for example test for bias towards the 3' end; if mRNA was sequenced via polyA selection, one would expect to see more sequences at the 3' end if the RNA was degraded. I'm confused why the authors wouldn't look at actual splicing events in the reads for the differential expression analysis, as they do later in the paper for the nucleotide divergence analysis. Just doing this analysis with genes for which there is hard evidence for splicing would be a lot more convincing.

2. There are two ways protein isoforms are made; through alternative splicing of the transcript, or through alternative promoters, when the gene is transcribed. Both result in differential exon expression, but they are entirely different biological processes. Calling differential exon expression 'alternative splicing' is very imprecise. The authors should either only look at genes with hard evidence for splicing (as mentioned in #1), or call it what it is; differential exon expression.

3. In the methods, the authors describe capturing exon expression independent of whole gene expression, but it is unclear how this affected the analysis in any way, or what the reader is supposed to interpret from this. I think the authors should explain better what this means.

4. The GO enrichment analysis is very vague. The authors state: "Enriched terms tended to cluster tightly in semantic space (Fig. 2D), suggesting that the set of genes affected exclusively by splicing has a narrow range of functions." It is unclear what 'semantic space' means, and I'm not sure I see the clustering the authors describe in figure 2D; to me they seem pretty well spread across the axes, and the different groups overlap a lot (although again, not sure what the axes mean). In the abstract the authors state: "many differentially spliced genes have unique functions", but I'm not sure the evidence supports that. I think the authors could better present and explain the data to support this conclusion.

5. The authors compare nucleotide divergence between coding sequences of genes. They find a consistently lower π in alternatively spliced genes. I think this result is really interesting. However, I think the authors should be very careful in how they interpret the result: nucleotide diversity or lack thereof within a coding sequence doesn't say anything about how a gene is regulated. I understand alternative splicing events are often cis regulated, but it is quite a big leap to conclude that reduced divergence in coding region means evidence for selection on cis regulatory processes. I think the authors should tune down their statement claiming to have a mechanism that 'drives regulatory cascades', and be very careful about suggesting to have found evidence of cis-regulatory evolution.

Minor edits:

line 102; though should be through

The SI figures S3-4 have overlapping text

Line 234; difficult to interpret, consider rewriting

Dear Dr. Tong and colleagues,

We were excited to see the enthusiastic comments and recommendations for improvement from the reviewers and appreciate the opportunity to revise our manuscript. Please find our point-by-point responses below. Note that all in-text citations below are also referenced in the main text, where their full citation can be found. We have also made several changes to comply with Nature Communications formatting guidelines, the largest of which was shortening the abstract, but also affected in-text citations and references to figures and tables in the supplementary materials.

Thank you for your time and input.

Sincerely,

RA Steward, MA de Jong, V Oostra, and CW Wheat

REVIEWER COMMENTS

Reviewer #1 (Remarks to the Author):

Phenotypic plasticity is a well known phenomenon where animals produce alternate phenotypes in response to changing environmental conditions using a single genotype. In this manuscript Steward et al. investigate the role of alternative splicing in seasonal plasticity in *Bicyclus anynana*, a textbook model of seasonal polyphenism. While previous reports in this system have investigated some physiological mechanisms underlying plastic wing pattern elements (Monteiro et al. 2015, Bhardwaj et al. 2018), little is known about alternative splicing modulating such plasticity. Steward et al. have analysed effects of seasonality and genetic background on plasticity in a lab population, and further expanded the study to include comparative study with population genomics study of a wild population.

Steward et al. found signatures of differential splicing across tissues, while few genes had seasonal splicing or expression. It was interesting to note that differentially expressed genes were not differentially spliced, which suggests both have a complementary non-redundant role to play in regulation of plasticity. Further they discuss constraints on evolution of plasticity in the natural population and lack of standing genetic variation in existing populations, as well as both cis-regulatory and trans environments as interesting candidates for future studies.

The work is excellent, experimental design neat and conclusions clear and concise. Supplementary materials were easy to follow through. The results are of significance to anyone working in the area of alternative splicing and phenotypic plasticity. I thoroughly enjoyed reading the manuscript and recommend it for publication.

We thank the reviewer for their positive feedback.

Suggestions :

Why were abdomen and thorax chosen as test samples to study plasticity and alternative splicing, when wings which show the most striking seasonal differences in the developmental life history of butterflies, excluded from the study? I believe that has to do with data availability at NCBI, but a discussion on relevance would be appreciated.

For this study, we used published, archived RNA-seq data available on NCBI. This data set was ideal because it had samples from multiple families and seasonal conditions, allowing us to test the genetic (G), environmental (E) and GxE variation in both splicing and gene expression in tissues from two body parts. While the wings might show the most striking developmental and morphological differences, the *Bicyclus anynana* polyphenism involves major physiological and reproductive differences between seasonal morphs. Hence, female abdomen and thorax tissues were chosen for the original work because they are critical tissues for this life history variation. There is certainly a role for splicing in the development of wing eyespots (Bear et al. 2017), but here our focus was on identifying transcriptional variation in adults and evaluating its link to genetic diversity.

To clarify our focus of study to the reader, we have added sentences to the introduction (lines 117 - 118; 125 - 126).

Did you find any sex-specific patterns of alternative splicing in these butterflies? It might additionally be worth discussing patterns of plasticity which is sex-limited and role reversal in alternate seasons (in context of Prudic et al. 2011).

This is a very interesting idea, especially given recent developments in the study of sex-specific splicing (e.g., Rogers et al. 2020, Agrawal and Singh 2021)! Unfortunately, the archived samples were exclusively from female butterflies (as described in the methods). To provide greater clarity for readers, we edited the introduction (lines 125 - 126) stating that the butterflies were all female. As stated above, we were primarily interested in life history traits (e.g., pace of life, reproductive investment) and these are most dramatic in females.

It might be out of the scope of this study, but I would be curious to know if any of the genes you found in this study overlapped with genes such as trehalase, cortex or herfst, as found in van der Berg et al., 2020.

We thought this was excellent idea! We added both doublesex and ecdysone receptor to this list of genes warranting a closer look, based on their roles in seasonal plasticity. We identified these genes in the *B. anynana* genome using our eggNOG functional annotation and confirmed their identities using BLASTn and sequences available on NCBI and LepBase. We did not find seasonal plasticity in splicing for any of these genes. In fact, only trehalase was differentially spliced in any of our analyses, but only among families in the abdomen (Table 1). While all these genes play a role in *B. anynana* transcriptional plasticity, the absence of a major role for splicing led us to conclude that discussing these genes would not add substantially to the manuscript. An interesting avenue for future research would be to investigate whether these genes are in networks that involve differentially spliced transcriptional regulators, but that is beyond the scope of this paper.

Table 1. Evidence of differential splicing or expression of genes known to be involved seasonal polyphenism and wing color patterning in Lepidoptera. None: genes met expression thresholds to be included in the analysis but were neither differentially spliced (DS, both exon expression and event expression) nor differentially expressed (DE). NA: expression thresholds were not met.

Gene : B. anynana locus	Abdomen			Thorax		
	Season	Family	SxF	Season	Family	SxF
trehalase : LOC112048234	DE	DE&DS	None	DE	None	None

cortex (cort): LOC112043008	DE	DE	None	NA	NA	NA
herfst/ CG8930 : LOC112049894	DE	DE	None	None	DE	None
doublesex (Dsx): LOC112058347	DE	DE	None	NA	NA	NA
Ecdysone receptor (EcR): LOC112043737	DE	None	None	DE	None	None

Edits :

Line 350 : Please change to nonsense mediated decay.

Line 384: corrected in the text

Line 602 : Grantham and Brisson (check comma)

Line 666: corrected in the text

Reviewer #3 (Remarks to the Author):

The authors investigate the role of alternative splicing in seasonal plasticity. They present a small subset of genes that are alternatively spliced between seasons, especially when family is taken into account. Testing for overlap between differentially expressed genes and differentially spliced genes suggested splicing as an alternative to differential gene expression. The authors show reduced genetic variation in genes with alternative splicing events.

Overall, I think the study of alternative splicing in seasonal plasticity is interesting. Investigating evidence of selection in genes associated with seasonal plasticity is especially important, as that topic has been relatively understudied.

We thank the reader for seeing the merits of our study.

I have some major concerns with the methodology and the presentation, that I think the authors should address.

1. The selection of alternatively spliced genes. If I understand correctly, alternative splicing was determined by differential exon expression within genes. This makes the assumption that genes were evenly sequenced across the transcript, an assumption that is not addressed at all. The authors could for example test for bias towards the 3' end; if mRNA was sequenced via polyA selection, one would expect to see more sequences at the 3' end if the RNA was degraded.

We agree with Rev. 3 that exon-based approaches like edgeR's diffSpliceDGE function are sensitive to both annotation quality and read coverage. We also agree that mRNA degradation is something to be concerned about. Our mRNA was sequenced via polyA selection and our samples do not suffer from high levels of degradation, which would bias coverage to the 3' end. While we have assessed this internally several years ago, we now document this for the reader, which should have been included in our MS. We have now added a test of read-coverage bias and a brief discussion of the consequences of such a bias to the Methods section (lines 472 - 478), which refers to a plot of the normalized coverage in the supplemental material (Supplementary Fig. 14).

I'm confused why the authors wouldn't look at actual splicing events in the reads for the differential expression analysis, as they do later in the paper for the nucleotide divergence

analysis. Just doing this analysis with genes for which there is hard evidence for splicing would be a lot more convincing.

The issue raised here is addressed just below (point 2).

2. There are two ways protein isoforms are made; through alternative splicing of the transcript, or through alternative promoters, when the gene is transcribed. Both result in differential exon expression, but they are entirely different biological processes. Calling differential exon expression 'alternative splicing' is very imprecise. The authors should either only look at genes with hard evidence for splicing (as mentioned in #1), or call it what it is; differential exon expression.

We agree with the reviewer that exon-based and event-based analyses capture different types of isoform variation. First, we have in fact conducted extensive event-based analyses that are concordant with the exon-based approaches, because we, like the reviewer, wanted to make sure exon-based analyses were capturing alternative splicing events. However, for the main analysis, we chose to report the exon-based results only because only exon-based analyses allowed us to perform a full-factorial analysis of season and family effects, which was our original goal (the event-based analyses with which we are familiar only allow for simple pairwise comparisons). We want this to be clear to the reader as well, so we have added a brief discussion of our motivation and possible drawbacks of focusing on differential exon expression in the first four sections of the results (lines 176 – 183). In these sections, we have also revised our text to use the term “differential exon expression” when we specifically refer to our results.

However, when discussing the implications of our results, we continue to use the term “differential splicing”. This is because our results using both the exon-based (Fig. 3) and event-based (Fig. 4) analyses convincingly support our conclusions about differential splicing. Nevertheless, in response to the reviewer’s concerns (point 1 above), we have also performed analyses they suggested, directly comparing the nucleotide diversity of genes with differentially spliced events between seasons with differentially expressed genes. We found the same pattern as we originally presented using exon-based differential splicing gene set, namely that nucleotide diversity is decreased in differentially spliced genes. We now refer to these additional analyses and results in the text (lines 392 - 394) and methods (lines 686 - 689), and the relevant figures have been added to the supplemental materials (Supplementary Fig. 12).

3. In the methods, the authors describe capturing exon expression independent of whole gene expression, but it is unclear how this affected the analysis in any way, or what the reader is supposed to interpret from this. I think the authors should explain better what this means.

We believe the reviewer is referring to our description of the methods used to conduct PCAs on exon expression data:

“To capture exon expression independently of whole gene expression we also calculated a TMM-residual by subtracting exon TMM values from the average TMM across all exons within a gene within an individual and compared these values using PCA.”

To clarify this, we have now added text to the methods (lines 493 – 499) to explain how and why these calculations were done, emphasizing that these values were only used to cluster

samples in PCA analyses. However, when using the exon-based and event-based differential splicing analyses, these tools are designed to account for this directly.

4. The GO enrichment analysis is very vague. The authors state: "Enriched terms tended to cluster tightly in semantic space (Fig. 2D), suggesting that the set of genes affected exclusively by splicing has a narrow range of functions." It is unclear what 'semantic space' means, and I'm not sure I see the clustering the authors describe in figure 2D; to me they seem pretty well spread across the axes, and the different groups overlap a lot (although again, not sure what the axes mean). In the abstract the authors state: "many differentially spliced genes have unique functions", but I'm not sure the evidence supports that. I think the authors could better present and explain the data to support this conclusion.

Our previous use of semantic similarity multidimensional scaling analyses was meant to demonstrate a stark pattern seen in the GO terms: that very few functional terms are shared among the sets of genes with differential exon expression, differential whole gene expression, or both. Upon reflection, we agree this could be presented in a more quantitative fashion. We have revised the main text to now present our results in a quantitative fashion using Euler diagrams (Venn-like diagrams in which circle and intersection sizes scale with the number of enriched GO terms). We characterize this pattern further in the supplementary materials (Supplementary Fig. 3-6) where we show each of the GO terms (with truncated descriptions, full descriptions can be found in Supplementary Table 8) and the degree to which they were enriched ($-\log_{10}(\text{p-value})$), for each category of expression. Both the results (lines 254-278) and methods (lines 550-579) have been rewritten to reflect these changes.

5. The authors compare nucleotide divergence between coding sequences of genes. They find a consistently lower pi in alternatively spliced genes. I think this result is really interesting. However, I think the authors should be very careful in how they interpret the result: nucleotide diversity or lack thereof within a coding sequence doesn't say anything about how a gene is regulated. I understand alternative splicing events are often cis regulated, but it is quite a big leap to conclude that reduced divergence in coding region means evidence for selection on cis regulatory processes. I think the authors should tune down their statement claiming to have a mechanism that 'drives regulatory cascades', and be very careful about suggesting to have found evidence of cis-regulatory evolution.

We agree with the reviewer that we need to be circumspect in our conclusions. While we believe our hypotheses are well supported across two levels of analysis, we have now removed any language that can be construed as causal throughout the text, with specific attention paid to the abstract and the final paragraph of the conclusions (427 – 440).

Minor edits:

line 102; though should be through

line 99: corrected in the text

The SI figures S3-4 have overlapping text

These figures have been replaced (Fig. S3-6), and all figures in the main text and supplement have been checked for overlapping text.

Line 234; difficult to interpret, consider rewriting

We have rewritten the line to describe this interesting pattern more clearly (lines 241 – 246).

Reviewers' Comments:

Reviewer #1:

None

Reviewer #3:

Remarks to the Author:

This is the second version I'm reviewing of this manuscript on the role of alternative splicing in seasonal plasticity. Previously, I expressed my concerns related to sequencing, the difference between alternative splicing and differential exon expression, and some concerns related to presentation and interpretation of the results.

I was happy to see the authors did a thorough job addressing all these concerns.

(1) Assumption of even sequencing. The authors provide additional information addressing sequencing bias across the entire gene, alleviation concerns of bias in the reads.

(2) Alternative splicing vs differential exon expression. The authors chose to continue to report exon-based results, as this would allow for a full factorial analysis. They do, however, offer additional explanation of their reasoning and potential drawbacks, as well as an additional analysis using splicing events. The additional information is very helpful, and addresses my concerns.

(3) Thank you for the clarifying text on TMM-residuals.

(4) GO enrichment analysis. The figures are much clearer now, thank you.

(5) Language related to evolution of alternative spliced genes. The authors have adjusted their conclusions both in the abstract and in the main text.

Overall, I think the paper is much improved, and the results are easier to interpret and better supported by the evidence presented in the paper. I think this work is really interesting, and this paper provides a clear roadmap for future researchers interested in investigating alternative splicing in phenotypic plasticity.

Reviewer #4:

Remarks to the Author:

Here I am reviewing "Alternative splicing in seasonal plasticity and the potential for adaptation to environmental change", a manuscript that has been previously reviewed. From my reading of the responses to reviewer comments in concert with the revised manuscript, it appears that the authors have sufficiently addressed the previous concerns. I have listed additional concerns below. Addressing these concerns would make the manuscript clearer.

Line 59: "In order to understand the tempo and mode of how plasticity evolves..."

I agree that mode might be addressed in this study, but tempo? I can think of a lot of other empirical approaches that address mutational rate, selection strength, and phenotypic change, but none of them are used here. Thus, the use of "mode" seems inaccurate in the context of this study. It seems more like the big picture question that the authors are answering deals with mode (what changes) and constraints (what is most likely to change) rather than the speed at which something changes.

Lines 99-112: I had to read this paragraph about 3 times to understand the argument, and now I'm pretty sure I didn't miss anything, it just needs to be written clearer. The authors are trying to set up the argument that "... alternately spliced genes may be more susceptible to erosion of genetic variation compared to modules [genes?] of differentially expressed genes." I have a couple gripes with this argument.

i. Is it really fair to compare individually spliced genes to groups of genes (modules)? I don't think that the authors do this kind of asymmetric comparison, but that is what they say in this paragraph, and it confuses me even before I see the experimental design. Why invoke modules vs. a direct comparison of just a DE gene to a DS gene?

ii. The authors argue trans-regulation is more important to differential gene expression, but cis-regulation is more important to differential splicing. Assuming this is true, why does this mean that

differentially spliced genes are more susceptible to winnowing of genetic variation? It's like a whole sentence connecting those two ideas was completely omitted.

This paragraph needs to rest on some proposed evolutionary mechanism for why we would expect reduced genetic variation in genes governed by cis- vs trans-regulatory mechanisms (assuming we could even draw a distinction between the two). The authors have provided no such mechanism here, requiring the readers to take a leap of faith regarding their predictions.

Line 114: Is it really necessary here to say that genetic variation for plasticity is depleted? I realize that this is based on a previous transcriptional study, but clearly it is not completely depleted as reaction norms in these butterflies clearly evolve, as shown in Van Bergen 2017 (not cited anywhere in this manuscript)

Line 202: Could you start this sentence out by saying "across tissues" or if that's not correct, "in the thorax/abdomen"? In general, it's hard in this paragraph to align the reported percentages with the specific comparisons being made. For instance, in the sentences – "We detected very few genes with among-family variation (SxF) in seasonal exon expression or whole gene expression. In the abdomen, 0.6% were alternatively differentially spliced while 0.1% were differentially expressed." – I think the authors are saying that, of genes that are differentially spliced or expressed between seasons, only 0.6% and 0.1% displayed family variation... however, as the second sentence is currently written, it seems like the authors are say that 0.6% and 0.1% were differentially regulated between seasons.

Line 227: To claim significance here, the authors should be reporting a chi square value (even if they do in the methods, they should be reporting it in the results to give their readers more confidence in the claim).

Lines 255-259: "There was very little functional overlap among sets of enriched GO terms (Fig. 2d; Supplementary Fig. 2d, 3-6): GO terms were especially unlikely to be shared between sets of genes with only differential whole gene expression and sets with only differential exon expression. Highly enriched GO terms were especially unlikely to be shared among sets of genes (Supplementary Fig. 3-6)."

These sentences seem partially redundant, and the second one is too vague to interpret.

Line 290: "Similarly, we detected increased π in whole genes that were differentially expressed between seasonal environments." This leads me to think that there was some difference in current analysis relative to the previous analysis (Oostra et al 2018); otherwise, why say "similarly" if it was just the same exact analysis on the same exact data. So, assuming there was a difference, that difference should be outlined here to distinguish it from the previous study.

Line 294: "However, in genes with differential exon expression between seasons, π was consistently lower... Interestingly, this reduced genetic variation in differentially spliced genes with differential exon expression was unique to the seasonal polyphenism." This is really confusing. You just defined them as genes at those with differential exon expression between seasons, so I'm not sure why it is further surprising/interesting that they are unique to seasonal polyphenism (they were defined that way)." Then the authors go on to say, "Among families, both genes with differential exon expression and those with differential whole gene expression were associated with statistically meaningful increases in π "... so differential expression between what? Just family variation? This was my same problem as I had earlier – using "differential" implies some kind of treatment or morph specific comparison, as opposed to family genetic variation which is assumed to be a distribution of effects. I'd highly recommend being more specific about "genes that exhibit family variation" rather than simply "differential" which easily gets confused with differentially expressed between seasonal morphs.

Line 313: Again, I feel like this statement: "...causal (likely cis-regulated) genes..." is unsubstantiated. The difference between cis- and trans-regulation is often blurred, and assuming you could make this distinction, there isn't substantial evidence that cis-regulated genes are any more causal than trans-regulated genes. I appreciate the distinction between causal and downstream genes, but don't understand the need to bring cis- and trans-regulation into the

paragraph at all.

Figure 3: The x-axis labels on these boxplots need to be redrawn... they are practically bumping into each other, making it hard to discern what grouping each box corresponds to (e.g., "not DE or DS DS only" looks like the same label)

Response to reviewers

Dear colleagues,

Thank you for investing your time and energy into reviewing our manuscript. We appreciate your help in strengthening our paper. Point-by-point responses to reviewer comments are below.

Warmly,

The authors

Reviewer #3 (Remarks to the Author):

This is the second version I'm reviewing of this manuscript on the role of alternative splicing in seasonal plasticity. Previously, I expressed my concerns related to sequencing, the difference between alternative splicing and differential exon expression, and some concerns related to presentation and interpretation of the results.

I was happy to see the authors did a thorough job addressing all these concerns.

1. Assumption of even sequencing. The authors provide additional information addressing sequencing bias across the entire gene, alleviation concerns of bias in the reads.
2. Alternative splicing vs differential exon expression. The authors chose to continue to report exon-based results, as this would allow for a full factorial analysis. They do, however, offer additional explanation of their reasoning and potential drawbacks, as well as an additional analysis using splicing events. The additional information is very helpful, and addresses my concerns.
3. Thank you for the clarifying text on TMM-residuals.
4. GO enrichment analysis. The figures are much clearer now, thank you.
5. Language related to evolution of alternative spliced genes. The authors have adjusted their conclusions both in the abstract and in the main text.

Overall, I think the paper is much improved, and the results are easier to interpret and better supported by the evidence presented in the paper. I think this work is really interesting, and this paper provides a clear roadmap for future researchers interested in investigating alternative splicing in phenotypic plasticity.

We were glad to hear that our revisions fully addressed Reviewer 3's concerns and we sincerely thank them for their time and effort.

Reviewer #4 (Remarks to the Author):

Here I am reviewing "Alternative splicing in seasonal plasticity and the potential for adaptation to environmental change", a manuscript that has been previously reviewed. From my reading of the responses to reviewer comments in concert with the revised manuscript, it appears that the authors have sufficiently addressed the previous concerns. I have listed additional concerns below. Addressing these concerns would make the manuscript clearer.

Line 59: “In order to understand the tempo and mode of how plasticity evolves...”

I agree that mode might be addressed in this study, but tempo? I can think of a lot of other empirical approaches that address mutational rate, selection strength, and phenotypic change, but none of them are used here. Thus, the use of “mode” seems inaccurate in the context of this study. It seems more like the big picture question that the authors are answering deals with mode (what changes) and constraints (what is most likely to change) rather than the speed at which something changes.

Line 47: We agree this could mislead some readers. We therefore removed both “tempo” and “mode”.

Lines 99-112: I had to read this paragraph about 3 times to understand the argument, and now I’m pretty sure I didn’t miss anything, it just needs to be written clearer. The authors are trying to set up the argument that “... alternately spliced genes may be more susceptible to erosion of genetic variation compared to modules [genes?] of differentially expressed genes.” I have a couple gripes with this argument.

i. Is it really fair to compare individually spliced genes to groups of genes (modules)? I don’t think that the authors do this kind of asymmetric comparison, but that is what they say in this paragraph, and it confuses me even before I see the experimental design. Why invoke modules vs. a direct comparison of just a DE gene to a DS gene?

ii. The authors argue trans-regulation is more important to differential gene expression, but cis-regulation is more important to differential splicing. Assuming this is true, why does this mean that differentially spliced genes are more susceptible to winnowing of genetic variation? It’s like a whole sentence connecting those two ideas was completely omitted.

This paragraph needs to rest on some proposed evolutionary mechanism for why we would expect reduced genetic variation in genes governed by cis- vs trans-regulatory mechanisms (assuming we could even draw a distinction between the two). The authors have provided no such mechanism here, requiring the readers to take a leap of faith regarding their predictions.

We appreciated the reviewer expressing their confusion here. We have revised the paragraph to focus on the differences between cis-regulatory motifs involved in splicing and whole gene expression variation and why these might have different impacts on the accumulation of genetic variation in the CDS. In this way, we provide a solid foundation for our predictions and avoid drawing a stark comparison between cis- and trans- regulatory mechanisms. (lines 89-107)

Line 114: Is it really necessary here to say that genetic variation for plasticity is depleted? I realize that this is based on a previous transcriptional study, but clearly it is not completely depleted as reaction norms in these butterflies clearly evolve, as shown in Van Bergen 2017 (not cited anywhere in this manuscript)

The previous study (Oostra et al. 2018) found genetic variation for transcriptional plasticity between seasons (GxE) in 1% of expressed genes. Furthermore, these genes had higher proportion of low-frequency polymorphisms (i.e., a more negative Tajima’s *D*) when compared with the remainder of the annotated genes, suggesting that a substantial number of polymorphisms in these GxE genes are under purifying selection and are being kept at low frequencies. Based on these two lines of evidence, we feel confident in our statement that genetic variation for plasticity is depleted.

As we understand their results, Van Bergen et al. (2017) found strong correlations among several traits involved in Mycalesine seasonal polyphenisms. Most of these correlations were conserved across species, as were the reaction norms of individual traits. However, in some cases, especially eyespots on the dorsal wing surface, correlations found within species were not shared among species, suggesting that these traits are “uncoupled” from the larger suite of seasonally plastic traits and evolving independently in these taxa. While we do believe this is strong evidence that the seasonal polyphenism has evolved (and may still be evolving) differently in these species, it does not inform on the levels of heritable variation within each of those species; each species could lack heritable variation in that reaction norm.

Nevertheless, we edited the sentence in the Introduction to state more clearly that “depleted genetic variation” refers to the Oostra et al. (2018) results. (Line 117). Also, because Van Bergen et al. (2017) perfectly capture the broad suite of correlated phenotypes that make up this polyphenism, we have cited this paper in our introduction of the suite of traits that differ among seasonal morphs (line 116).

Line 202: Could you start this sentence out by saying “across tissues” or if that’s not correct, “in the thorax/abdomen”? In general, it’s hard in this paragraph to align the reported percentages with the specific comparisons being made. For instance, in the sentences – “We detected very few genes with among-family variation (SxF) in seasonal exon expression or whole gene expression. In the abdomen, 0.6% were alternatively differentially spliced while 0.1% were differentially expressed.” – I think the authors are saying that, of genes that are differentially spliced or expressed between seasons, only 0.6% and 0.1% displayed family variation... however, as the second sentence is currently written, it seems like the authors are saying that 0.6% and 0.1% were differentially regulated between seasons.

In this paragraph, we primarily focus the text on the results for the abdomen, whereas the thorax results are only reported in the figures and supplement. Thus, all percentages in this paragraph describe patterns in the abdomen. We have added text in line 199 clarifying this. In addition, we combined the sentences describing the number of genes with SxF effects to emphasize that these percentages refer to the number of genes with differential splicing and/or expression. (lines 204 – 210)

Line 227: To claim significance here, the authors should be reporting a chi square value (even if they do in the methods, they should be reporting it in the results to give their readers more confidence in the claim).

To calculate significant overlap here, we used Fisher’s Exact Tests for count data on 2x2 contingency tables. In these cases, the p-value is calculated directly using the hypergeometric distribution based on the odds ratio. There is no X^2 value or F statistic to report, only the odds-ratios and p-values already reported in Supplementary table 7. We have revised the caption of this table to better help readers understand these results. We also revised the main text, adding the relevant odds ratio and p-value (lines 226 -227)

Lines 255-259: “There was very little functional overlap among sets of enriched GO terms (Fig. 2d; Supplementary Fig. 2d, 3-6): GO terms were especially unlikely to be shared between sets of genes with only differential whole gene expression and sets with only differential exon expression. Highly enriched GO terms were especially unlikely to be shared among sets of genes (Supplementary Fig. 3-6).”

These sentences seem partially redundant, and the second one is too vague to interpret.

We were interested to observe that several GO terms were shared among gene sets, but that these were never the most highly enriched terms. However, it is clear from the reviewer's comments that our explanation of this result detracts from the message of the paragraph. We decided to remove the sentence from the text. (lines 253 -254)

Line 290: "Similarly, we detected increased pi in whole genes that were differentially expressed between seasonal environments." This leads me to think that there was some difference in current analysis relative to the previous analysis (Oostra et al 2018); otherwise, why say "similarly" if it was just the same exact analysis on the same exact data. So, assuming there was a difference, that difference should be outlined here to distinguish it from the previous study.

We have added text reminding the reader of the differences between the analyses in the two papers, namely, the population genetic data were calculated from a sample of wild *B. anynana* butterflies, whereas Oostra et. al. (2018) calculated population genetic parameters directly from the RNA data. While the RNA data are the same, they were mapped to different references (previously, a transcriptome assembled from these data, and for ours, the annotated reference genome available on NCBI) and filtered and normalized using different techniques that are part of the edgeR analysis pipeline. We also restricted our analyses to multiexon genes, so that we could better compare how splicing and whole gene expression interact. (lines 277 – 279).

Line 294: "However, in genes with differential exon expression between seasons, pi was consistently lower... Interestingly, this reduced genetic variation in differentially spliced genes with differential exon expression was unique to the seasonal polyphenism." This is really confusing. You just defined them as genes at those with differential exon expression between seasons, so I'm not sure why it is further surprising/interesting that they are unique to seasonal polyphenism (they were defined that way)." Then the authors go on to say, "Among families, both genes with differential exon expression and those with differential whole gene expression were associated with statistically meaningful increases in pi"... so differential expression between what? Just family variation? This was my same problem as I had earlier – using "differential" implies some kind of treatment or morph specific comparison, as opposed to family genetic variation which is assumed to be a distribution of effects. I'd highly recommend being more specific about "genes that exhibit family variation" rather than simply "differential" which easily gets confused with differentially expressed between seasonal morphs.

We understand that the design of the study complicates our discussion of the results, and we have managed this complexity in some places better than others. Here, we were trying to point out that the pattern of reduced pi in seasonally spliced genes was not detected in genes that were differentially spliced among families, suggesting that decreased pi is not a consequence of alternative splicing itself. We have edited the paragraph to make this observation clearer. (Lines 282 -288)

We disagree with the suggestion that variation in whole gene expression among families should not be described as "differential expression". Numerous contemporary studies examine differential expression among genotypes, families, generations, strains, populations,

and species, all as proxies for genetic variation in gene expression (e.g., Andrew et al. 2021; Bittner et al. 2021; Hamann et al. 2021; Hsu et al. 2021). We have tried to consistently refer to these effects as “among-family differential [whole gene/ exon] expression” or “differential [whole gene/ exon] expression among families” throughout the paper.

Line 313: Again, I feel like this statement: “...causal (likely cis-regulated) genes...” is unsubstantiated. The difference between cis- and trans-regulation is often blurred, and assuming you could make this distinction, there isn’t substantial evidence that cis-regulated genes are any more causal than trans-regulated genes. I appreciate the distinction between causal and downstream genes, but don’t understand the need to bring cis- and trans-regulation into the paragraph at all.

In line with the changes we made in the introduction, we walked back our discussion of cis- and trans- regulation here, instead focusing on differential expression networks and how we hypothesize the signature of strong selection on environmentally-sensitive regulatory genes dissipates across a regulatory cascade. As our text emphasizes, this explanation is hypothetical, and we hope to explore it further in future work. (Lines 303 – 308)

Figure 3: The x-axes labels on these boxplots need to be redrawn... they are practically bumping into each other, making it hard to discern what grouping each box corresponds to (e.g., “not DE or DS DS only” looks like the same label)

We were happy to fix this and believe it is now clearer and more legible.

References

- Andrew SC, Primmer CR, Debes PV, Erkinaro J, Verta J-P. 2021. The Atlantic salmon whole blood transcriptome and how it relates to major locus maturation genotypes and other tissues. *Marine Genomics* 56:100809.
- van Bergen E, Osbaldeston D, Kodandaramaiah U, Brattström O, Aduse-Poku K, Brakefield PM. 2017. Conserved patterns of integrated developmental plasticity in a group of polyphenic tropical butterflies. *BMC Evolutionary Biology* 17:59.
- Bittner NKJ, Mack KL, Nachman MW. 2021. Gene expression plasticity and desert adaptation in house mice*. *Evolution* 75:1477–1491.
- Hamann E, Pauli CS, Joly-Lopez Z, Groen SC, Rest JS, Kane NC, Purugganan MD, Franks SJ. 2021. Rapid evolutionary changes in gene expression in response to climate fluctuations. *Molecular Ecology* 30:193–206.
- Hsu S-K, Belmouaden C, Nolte V, Schlötterer C. 2021. Parallel gene expression evolution in natural and laboratory evolved populations. *Molecular Ecology* 30:884–894.
- Oostra V, Saastamoinen M, Zwaan BJ, Wheat CW. 2018. Strong phenotypic plasticity limits potential for evolutionary responses to climate change. *Nat Commun* 9:1–11.